# Synthesis, Characterization and Evaluation of a Novel Tetraphenolic Compound as a Potential Antioxidant

**DOI:** 10.3390/antiox12071473

**Published:** 2023-07-22

**Authors:** Mengqi Xu, Pengcheng Meng, Hongyan Wang, Jun Liu, Tao Guo, Zhenjie Zhu, Yanlan Bi

**Affiliations:** 1College of Food Science and Technology, Henan University of Technology, Zhengzhou 450001, China; 2021920059@stu.haut.edu.cn (M.X.); pengchengmeng@haut.edu.cn (P.M.); liujun@haut.edu.cn (J.L.); 2021920086@stu.haut.edu.cn (Z.Z.); 2Food Laboratory of Zhongyuan, Luohe 462300, China; 3College of Chemistry and Chemical Engineering, Henan University of Technology, Zhengzhou 450001, China; why@haut.edu.cn (H.W.); taoguo@haut.edu.cn (T.G.)

**Keywords:** tetrahydroxyl-phenol antioxidants, synthesis and purification, free radical scavenging, lipid antioxidant

## Abstract

A novel antioxidant containing four hydroxyl groups, namely 2,2′-(2-methylpropane-1,3-diyl)bis(hydroquinone) (MPBHQ), was synthesized using hydroquinone and methylallyl alcohol as the raw materials, phosphoric acid as the catalyst, and toluene as the solvent system. The structure of MPBHQ was characterized by mass spectrometry, nuclear magnetic resonance, ultraviolet spectroscopy, and infrared spectroscopy. The results showed that MPBHQ has a good radical scavenging effect, as measured by the ORAC assay, DPPH radical scavenging assay, ABST radical scavenging assay, and Rancimat test. In fatty acid methyl ester and lard without exogenous antioxidants, MPBHQ showed better antioxidant performance than butylated hydroxytoluene (BHT), hydroquinone (HQ), tert-butyl hydroquinone (TBHQ), and propyl gallate (PG), meeting the need for a new antioxidant with better properties to ensure the oxidative stability of lipids and biodiesel.

## 1. Introduction

Phenolic antioxidants are the main types of antioxidants, most of which are hydroxyl groups directly linked to the aromatic ring [1]. These compounds can provide hydrogen protons to block free-radical-dominated chain reactions [2], which gives them good antioxidant activity in oil and fat products and other chemical products. Although consumers currently prefer natural phenolic antioxidants, such as tocopherols, tea polyphenols, syringic acid, and rosmarinic acid [3], they are usually expensive due to their restricted sources and difficulty in purification [4]. Moreover, they have undesirable flavors for specific products or people [5]. The mainstream of the market is still synthetic phenolic antioxidants, which are widely used in food [1], medicine [6], cosmetics [7], fuel [8], feed [9], and other fields [10], by virtue of their low price, high yield, and good antioxidant effect.

Synthetic phenolic antioxidants have a similar structure, which is usually a phenolic hydroxyl group and an alkyl group attached to the benzene ring [11]. Commercial synthetic phenolic antioxidants, including butyl hydroxyanisole (BHA), butylated hydroxytoluene (BHT), and tert-butyl hydroquinone (TBHQ), are commonly used to inhibit the oxidative degradation of plastics and rubber and to delay the oxidative rancidity of food when used as food additives [12,13]. TBHQ is a common synthetic antioxidant containing two hydroxyl groups, which has a significant antioxidant effect on vegetable oils with high unsaturated fatty acids [14,15]. Propyl gallate (PG) is also a common synthetic polyphenol antioxidant, which is commonly used in oil-rich foods, such as fried foods and cookies [16]. However, most of the existing synthetic phenolic antioxidants are usually volatile at high temperatures (frying) and are prone to decomposition during the storage of oil and fat products [17], which is mainly because of their small molecular weight [18], and these defects reduce their antioxidant properties. Moreover, the application of synthetic phenolic antioxidants mainly focuses on mono-hydroxy and di-hydroxy species [19], and there is a lack of research on and application of synthetic polyphenolic antioxidants. Therefore, new antioxidants of a larger molecular weight with excellent thermostability are favored, and they may have better antioxidant properties and more extended application fields.

Hydroquinone (HQ) is an important raw material additive and intermediate in the synthesis of pharmaceuticals [20] and other fine chemicals. But HQ is less used in the oil and fat processing industry due to its poor oil solubility, small molecular weight, and easy volatility [21]. For the synthesis of phenolic antioxidants, HQ is an important reactant for the synthesis of TBHQ. TBHQ is synthesized from hydroquinone and tert-butyl alcohol by the Foucault alkylation reaction, which enhances the oil solubility and the antioxidant capacity of a synthetic antioxidant. It is well-known that the antioxidant capacity of natural phenolic antioxidants containing four phenolic hydroxyl groups, such as epicatechin and rosmarinic acid, is significantly stronger than that of tocopherol containing one hydroxyl group and TBHQ containing two phenolic hydroxyl groups [11,22,23]. However, TBHQ has a molecular weight of 164 and only contains two phenolic hydroxyl groups, which leads to its limited antioxidant capacity and the universality of the use environment [17]. Based on this design theory, 2-(*tert*-butyl)-5-methylbenzene-1,4-diol (TBMHQ) containing four phenolic hydroxyl groups was synthesized [24], and the DPPH radical scavenging test and oxidative decay test results showed that it has enhanced antioxidant properties. Therefore, thermal stability and antioxidant activity can be improved by increasing the molecular weight of the substance and introducing more hydroxyl groups. In addition, the introduction of alkyl terminations improves the oil solubility of the antioxidant. The alkylating reagent chosen is methyl allyl alcohol, which has one more double bond than the usual alkylating reagent used in the synthesis of antioxidants [25]. It is designed to increase the molecular weight of hydroquinone by the Friedel–Crafts alkylation reaction.

In this work, a novel synthetic polyphenol antioxidant named 2,2′-(2-methylpropane-1,3-diyl)bis(hydroquinone) (MPBHQ) containing four hydroxyl groups was successfully synthesized based on the Friedel–Crafts reaction. The molecular weight and main chemical properties of MPBHQ were identified by high-resolution mass spectrum (HRMS), retention factor (Rf) value, and UV spectroscopy, and the chemical structure was also confirmed by HRMS, nuclear magnetic resonance (NMR), and Fourier-transform infrared (FTIR) spectroscopy. The antioxidant properties of MPBHQ were studied by the ORAC assay, DPPH radical scavenging assay, ABTS radical scavenging assay, and Rancimat test. MPBHQ exhibits strong antioxidant activity in its edible oil system and mixed fatty acid methyl ester. Therefore, MPBHQ, a novel synthetic antioxidant with a definite chemical structure and excellent antioxidant capacity, has broad application prospects in the food and chemical industries.

## 2. Materials and Methods

### 2.1. Materials

Toluene, hydroquinone, 2-methylallyl alcohol, phosphoric acid, potassium persulfate, C18 column chromatography silica gel (40–60 μm), iodine (AR), and other non-specifically indicated chemicals of analytical purity grade were from Shanghai Aladdin Biochemical Technology Co., Ltd. (Shanghai, China). Silica-gel-coated plates (100 × 100 mm, 0.25 mm) were purchased from Qingdao Ocean Chemical Co., Ltd. (Qingdao, China). ABTS (>98%), DPPH (>98%), mixed fatty acid methyl ester (fatty acid methyl ester, ≥99%), 2,2′-azobis(2-amidinopropane) dihydrochloride (AAPH), Trolox (6-hydroxy-2,5,7,8-tetramethylchroman-2-carboxylic acid), which is a hydrophilic analogue of vitamin E, fluorescein sodium salt, tert-butylhydroquinone (TBHQ, >98%), butylated hydroxytoluene (BHT, >98%), and propyl gallate (PG, >98%) were also purchased from Shanghai Aladdin Biochemical Technology Co., Ltd. Refined lard without antioxidants was purchased from Tianjin Jiuyuan Oils and Fats Co., Ltd. (Tianjin, China). Chromatographically pure methanol was obtained from VBS Biologic Inc. (New York, NY, USA). Deuterated methanol (≥99.8%) was obtained from Cambridge Isotope Laboratories, Inc. (Andover, MA, USA).

### 2.2. Synthesis of MPBHQ

MPBHQ was synthesized through a liquid-phase reaction strategy using phosphoric acid as the catalyst. The main factors, including reaction temperature, reaction time, and reactant ratio, were chosen to investigate the effect on the synthesis of MPBHQ. Typically, 4.4 g of hydroquinone and 4 mL of phosphoric acid were added to 20 mL of toluene with magnetic stirring in a thermostatic water bath. When the reaction temperature reached 90 °C, 4 mL of 2-methylallyl alcohol was added dropwise to the above mixture. After the reaction was performed for 20 min at 90 °C with magnetic stirring, the reaction products were transferred into a pear-shaped separatory funnel. Then, the aqueous phase containing phosphoric acid was separated, and the unreacted p-phenol was also separated by washing with hot water. The retained organic phase was transferred into a 100 mL round bottom flask, and the toluene was evaporated at 70 °C using a rotary evaporator. Eventually, a white powder was obtained after drying in a vacuum oven at 70 °C for 6 h. The synthesis of MPBHQ and the harvest process of the crude products are shown in Figure 1.

### 2.3. Content of MPBHQ

The content of MPBHQ was analyzed by high-performance liquid chromatography (HPLC). The sample was dissolved in methanol, and this solution was purified by a 0.45 µm filter membrane. The sample concentration was finally fixed at 1 mg·mL^−1^. The signal of the compound was collected on a high-performance liquid chromatograph (Waters 2695, Waters Corp., Milford, MA, USA) equipped with a C18 column (Waters SunFire, 4.6 mm × 250 mm, 5 µm) and an ultraviolet (UV)/visible detector (Waters 2489, Waters Corp., Milford, MA, USA). The detection wavelength was 280 nm, the column temperature was 35 °C, and the injection volume was 20 µL. The mobile phase was a mixture of methanol and water, and the phase A was aqueous solution containing 0.5% acetic acid, phase B was methanol. The flow rate of the mobile phase was 0.8 mL·min^−1^. The samples were analyzed according to the following gradient procedure. The elution profile was as follows: 0 min, 60% A (aqueous solution containing 0.5% acetic acid); 1–8 min, 60–20% A; 8–15 min, 20–0% A; 15–20 min, 0% A; 20–28 min, 0–60% A. The content of MPBHQ was calculated according to the area normalization method.

### 2.4. Separation and Purification

Reversed-phase column chromatography coupled with a laboratory self-loaded C18 column (17 × 300 mm, 40–60 μm) was selected for the separation and purification of the crude products containing MPBHQ. After the C18 silica-gel was activated by using methanol for 8 h, it was loaded using a wet-process column-loading strategy. Before the separation process, the column was kept flat with quartz sand and equilibrated with the eluent until the column bed was uniform and bubble-free. After dissolving the crude products containing MPBHQ in methanol completely, the solution was loaded in the pre-column and separated in the following elution program. The mobile phase was a mixture of methanol and water (methanol:water, 4:6, *v*/*v*), the flow rate was 5 mL·min^−1^, and the sample loading mass was 0.2 g. Finally, the eluate was collected into a 10 mL test tube for subsequent analysis.

### 2.5. Identification of the Structure of MPBHQ

The structure of MPBHQ was identified based on the information from the HRMS, NMR, and FTIR. The chemical bond information of MPBHQ was recorded from a Brucker ALPHA FTIR spectrometer (Bruker, Karlsruhe, Germany) using ATR with the wavenumber range of 400–4000 cm^−1^. For LC-MS analysis, the conditions for this part of the liquid chromatography (LC) were the same as those in Section 2.3., but the injection volume was 10 µL. The ionization of substances depended on the Electron Spray Ionization (ESI). The executive mass spectrometer was performed in the negative ion mode. The mass range was 110–500 *m*/*z*. The high-purity nitrogen was used as a nebulizer and drying gas, and the desolvation gas flow rate was 380 L/h. The ToF-MS functions were set as follows: the desolvation temperature was 180 °C, the source temperature was 80 °C, the capillary voltage was 2.9 kV, and the sampling cone voltage was 30 eV. ^1^H NMR, ^13^C NMR, 2D COSY, 2D HSQC, and 2D HMBC spectra were recorded at 25 °C on a Bruker AVANCE III 500 MHz NMR spectrometer (500.13 MHz proton frequency) equipped with a BBI liquid probe and Z-gradient system. Deuterated methanol was used as a lock solvent, and the concentration of the working solution for NMR was 30 mg·mL^−1^. All the pulse programs used for NMR experiments were the standard ones equipped in Bruker software Topspin 3.2. For 1D−^1^H NMR and 1D−^13^C NMR spectra, and both of the pulse programs “zg” and “zgpg30” were used with a delay time of 2.0 s. The 2D COSY spectra were acquired with the pulse program “cosygpmfqf” with a relaxation delay of 2.0 s. A data matrix of 2048 × 128 points was recorded with 8 scans for each increment. Two-dimensional HSQC spectra were acquired with the pulse program “hsqcetgpsisp2.2” with a relaxation delay of 2.0 s and a J(CH) constant of 145 Hz. Two-dimensional HMBC spectra were acquired with the pulse program “hmbcgpndqf” with a relaxation delay of 1.5 s and a J(XH) constant of 8 Hz. The data matrix of 1024 × 256 points were recorded in HSQC and HMBC with 8 scans for each increment. ^1^H NMR (500 MHz, Methyl alcohol-d) δ 6.55 (d, J = 1.9 Hz, 1H, H-3), 6.35 (d, J = 2.0 Hz, 1H, H-1), 0.90 (d, 3H, H-6), 2.52 (m, H, H-5), 3.98 ppm (d, 2H, H-4). ^13^C NMR (500 MHz, Methyl alcohol-d) δ (118.2, C-5), δ (116.4, C-3), δ (115.2, C-2), δ (23.6, C-9), δ (33.6, C-8), δ (46.2, C-7). HRMS (ESI): Calcd for C_16_H_18_O_4_^−^ [M-H]^−^ *m*/*z* 273.0934, found 273.0934. The purity of MPBHQ sample was determined by HPLC as 98.0%.

### 2.6. Retention Factor Value

The retention factor (Rf) value of selected compounds was determined by thin layer chromatography (TLC). Typically, 5 μL of methanol solution containing 0.15 mg of the samples were spotted on a silica-gel plate. Then, the plate was developed in the chloroform methanol mixture (19:1, *v*/*v*), and the total expansion distance of TLC plate was 9 cm. After the expansion was completed and the solvent was evaporated, iodine was used to develop the color. The Rf value was calculated based on the shift distance of the sample and using the equation:Rf = d_c_/d_o_
where d_c_ and d_o_ represent the distance moved from the origin by component and the distance moved from the origin by solvent, respectively.

### 2.7. Free-Radical Scavenging Assay

DPPH radical scavenging rate and ABTS radical scavenging rate were selected as two indexes to evaluate the antioxidant capacity of MPBHQ. The DPPH radical scavenging assay of BHT, HQ, TBHQ, PG, and MPBHQ was performed as described in Jiang et al. [19]. Briefly, 0.5 mL of antioxidant methanol solution containing 1.5, 3.0, 6.0, 12.0, 24.0, and 48.0 µg·mL^−1^ was mixed with 3.0 mL of 0.1 mM DPPH methanol solution, respectively. This mixture was sufficiently shaken and reacted under darkness at room temperature for 30 min. The absorption was measured on a TU-1810 UV-visible spectrophotometer (PuXi Tongyong, Beijing, China) at 517 nm. The methanol was used as the blank test sample. A total of 0.5 mL of methanol was added in 3.0 mL DPPH solution, and this mixture served as the control.

ABTS radical scavenging activity of MPBHQ BHT, HQ, TBHQ and PG was determined according to the previous method reported by Feng [26]. A total of 9 mL of 7 mM ABTS aqueous solution was mixed with 1 mL of 24.5 mM potassium persulfate solution, and this mixture was diluted 50 times after it was left to react in the dark at room temperature for 16 h. Then, 0.2 mL of each antioxidant methanol solution was dispersed in 4.0 mL of ABTS solution and reacted in the dark for 6 min. The absorbance value was collected on a TU-1810 UV-visible spectrophotometer at 734 nm. Methanol was used as the blank test sample, and the mixture containing 0.2 mL of methanol and 4.0 mL of DPPH solution served as the control.

The radical scavenging activity (RSA) was calculated using the equation:RSA (%) = [(A_2_ − A_1_)/A_2_] ×100%
where A_1_ is the absorbance value of the tested sample, and A_2_ is the absorbance value of control. The free radical scavenging activity of antioxidants is expressed as EC_50_, which represents the effective concentration required to achieve 50% antioxidant capacity [27].

### 2.8. ORAC Assay

The oxygen radical absorption capacity (ORAC) assay was measured by a modified method, and this method used an enzyme marker (SPARK, Tecan, Männedorf, Switzerland) with an excitation wavelength of 485 nm and an emission wavelength of 530 nm. In each well, 150 μL of FL (78 nM) and 25 μL of sample, blank (PBS), or standard (Trolox, 6.25–100 μmol/L) were placed, and then 25 μL of AAPH (73 mM) were added. The fluorescence was measured immediately after the addition, and the measurements were then taken every 5 min until the relative fluorescence intensity (FI%) was less than 5% of the value of the initial reading. The measurements were taken in triplicate. The area under the fluorescence decay curve (AUC) was calculated by applying the following equations:ORAC (μm TE) = (C_Trolox_/C_sample_)·[(AUC_sample_ − AUC_blank_)/(AUC_Trolox_ − AUC_blank_)]
AUC = 0.5 × [2 × (f_0_ + f_1_ + … + f_n−1_ + f_n_) − f_0_ − f_n_]△t
where f_0_ is the initial fluorescence and f_n_ is the fluorescence at time n.

### 2.9. Rancimat Test

The antioxidant activities of MPBHQ, BHT, HQ, TBHQ, and PG in lard system and mixed fatty acid methyl ester system were determined according to the AOCS Official Method Cd 12b-92 Oil Stability Index, and the data were collected on a Rancimat 892 Apparatus (Metrohm, Herisau, Switzerland). A total of 3.00 ± 0.01 g of lard containing antioxidants were subjected to accelerated oxidation under the following conditions. The heating temperature was 120, 130, and 140 °C with an airflow rate fixed at 20 L·h^−1^, and the measuring cell distilled water dosage was 60 mL. The oxidation induction time of BHT, HQ, TBHQ, PG, and MPBHQ on lard at different concentrations (0.01%, 0.02%, 0.04%) was determined, and lard without antioxidant was the control. The protection factor (Pf) values were calculated following the equation:Pf = IP_sample_/IP_control_
where IP_sample_ and IP_control_ represent the induction period (IP) of lard samples with and without antioxidants, respectively.

### 2.10. Statistical Analysis

All experiments were performed in duplicate, and data were presented as mean ± standard deviation (SD). The data were analyzed by Duncan’s multiple range tests using ver. 26.0 SPSS (IBM SPSS, Almonk, NY, USA).

## 3. Results and Discussion

### 3.1. Synthesis, Purification and Characterization of MPBHQ

The novel tetra phenolic antioxidant MPBHQ was obtained as illustrated in Figure 1. The synthesis of MPBHQ was based on the phosphoric acid (H_3_PO_4_) catalyzed Friedel–Crafts alkylation of hydroquinone (A) and 2-methylallyl alcohol (B). Toluene as the solvent provided the reaction environment. The composition of the crude products containing MPBHQ were analyzed, and the possible products were shown in Figure 1a. The effect of reaction temperature, molar ratio of hydroquinone to methyl allyl alcohol, and reaction time on the synthesis of MPBHQ was investigated, and the results were shown in Table 1. When the molar ratio of hydroquinone, methyl allyl alcohol and H_3_PO_4_ was 1:1.2:0.5, the reaction temperature was 90 °C, and the reaction time was 20 min, the content of MPBHQ reached 41%. After the separation and purification according to the steps shown in Section 2.4 and Figure 1, the MPBHQ content in the purified product can reach 98.0%, which was determined by a high-performance liquid chromatograph.

The properties and structure of MPBHQ were identified and confirmed by TLC, ultraviolet spectroscopy, NMR, HRMS (ESI), FTIR, etc. For the Rf values of HQ and MPBHQ obtained by TLC, and Rf value can reflect the polarity of the substance [25]. The Rf values of HQ and MPBHQ were 0.20 and 0.45, respectively, which indicated that MPBHQ exhibited stronger lipophilicity than HQ. From (Figure 1b), MPBHQ showed a strong UV absorption peak centered at 297 nm. After mixing MPBHQ with KOH solution, the absorption peak is enhanced and red-shifted, indicating that MPBHQ is a phenolic compound. The chemical structure of MPBHQ was further confirmed by FTIR. From Figure 1c, the absorption peak centered at 3316.88 cm^−1^ is assigned to the stretching vibration of the phenolic hydroxyl group on the benzene ring [28]. The peaks located at 2949.95 and 1020.42 cm^−1^ are assigned to the stretching vibration and in-plane bending vibration of C-H on the benzene ring, respectively. The peaks centered at 2947.88 and 2834.50 cm^−1^ are attributed to the stretching vibration of C-H on alkanes, and at 1422.41 cm^−1^ is attributed to the bending vibrations of C-H on alkanes. In order to further determine the structure of the MPBHQ, its molecular weight was determined by HRMS. From Figure 2a, the primary-ion mass spectra of MPBHQ showed the fundamental peak *m*/*z* 273.0934 [M-H]^−^, which inferred its molecular weight to be 274. As shown in Figure 2b, the secondary-ion mass spectra of MPBHQ shows more detailed structural information, such as the benzene ring structure. The chemical structure of MPBHQ was identified by NMR spectra. The NMR spectra of MPBHQ was shown in Figure 3. As shown in Figure 3a, the chemical shifts of hydrogen on the benzene ring were located at around 6–7 ppm. The hydrogen on C9 has the biggest electron shielding effect in the molecule structure, so it shows the smallest chemical shift in the high field region at 0.90 ppm. The chemical shifts of hydrogen at H4 and H5 are 3.98 ppm and 2.52 ppm, respectively.

All the assignments of hydrogen chemical shifts in Figure 3a were further determined by the COSY spectra (Figure 3c). The information on the coupling of each group was obtained from the relevant peaks outside the diagonal of the COSY spectra. We can easily obtain the coupling information of H5 and H6 for H4 and H5. Based on the correlation signals of C-H in HSQC spectra (Figure 3d), all the assignments of carbons in MPBHQ molecules were shown in Figure 3b. The chemical shifts of carbons connecting the two phenolic hydroxyl positions (C1 and C4) are determined at 150.9 ppm and 149.5 ppm, respectively. The chemical shifts of carbons attached to hydrogens on the benzene ring were determined at the range of 110–120 ppm. For the alkyl fraction, the chemical shift of the methyl carbon was determined at 23.6 ppm. The C8 of methylene was assigned to the chemical shift of 33.6 ppm. Then, from Figure 3d, we found a correlation between the three hydrogen atoms on the benzene ring except the hydrogen atoms on the hydroxyl group and the carbon atoms of the benzene ring, which are H1, H2, and H3. The correlations of hydrogen atoms H4, H5, and H6 on alkyl corresponding to C7, C8, and C9 are also marked in the figures. In Figure 3e, the correlations of H and C in adjacent or interphase positions are marked, such as the correlations of H6 and C7, and H1 and C5. In summary, the chemical information of MPBHQ was determined, and it is an alkylbenzene compound containing four hydroxyl groups, as shown in Figure 3.

### 3.2. DPPH Radical Scavenging Assay

The DPPH radical scavenging assay has the advantages of being efficient, convenient, sensitive, and reproducible. Moreover, the EC_50_ value of the sample concentration corresponding to a free radical scavenging rate of 50% was calculated to reflect the scavenging efficiency of the antioxidants on free radicals [29]. MPBHQ contains four phenolic hydroxyl groups, which is presumed to have a strong antioxidant effect. MPBHQ and the compounds containing different amounts of phenolic hydroxyl groups, BHT, HQ, TBHQ, and PG, were selected to examine their scavenging ability against DPPH radicals, and the antioxidant capacity of MPBHQ was reflected by comparing the DPPH free radical scavenging rate. From Figure 4a, the DPPH radical scavenging rate of MPBHQ and other antioxidants increased with the increase in antioxidant concentrations. EC_50_ values of TBHQ, HQ, PG, and MPBHQ were 22.20, 11.34, 8.74, and 7.93 μg/mL, respectively. Due to the weak ability of BHT, the EC_50_ values of BHT cannot be calculated. Among all of the selected compounds, the DPPH radical scavenging ability of HQ, PG, and MPBHQ increased rapidly with increasing concentration from 1.5 to 24 μg/mL. Moreover, when the compound concentration was 24 μg/mL, their free radical scavenging rate reached 86.10%, 86.21%, and 93.85%, respectively. Overall, the DPPH radical scavenging ability was MPBHQ > PG > HQ > TBHQ > BHT. That is to say, the more phenolic hydroxyl groups of the compound, the better the scavenging activity on DPPH radicals. The scavenging ability of MPBHQ containing tetrahydroxyl groups was almost twice as high as that of TBHQ. This is mainly because of the enhanced hydrogen supply capacity from the presence of two hydroquinones in the structure and more DPPH capture by the released [H] from the stable resonators [18]. MPBHQ combined the hydrogen donating capacity from the two hydroquinones. Therefore, the DPPH radical scavenging capacity is almost twice that of TBHQ. Based on the selected method in this work, the DPPH radical scavenging assay in methanol solution is mainly dominated by single electron transfer (SET) mechanism. However, ABTS radicals can be eliminated by both SET and hydrogen atom transfer (HAT) mechanism. In order to evaluate the antioxidant activity of MPBHQ in detail, ABTS radical scavenging assay was carried out.

### 3.3. ABTS Radical Scavenging Assay

The ABTS radical scavenging activity of the selected antioxidant compounds was shown in Figure 4b. The scavenging rate of ABTS radicals increased with the increase in the antioxidant concentration (1.5–48 μg/mL). EC_50_ values of TBHQ, MPBHQ, HQ, and PG were 33.34, 24.35, 21.81, and 18.17 μg/mL, respectively. BHT still showed weak scavenging activity, and the scavenging ability of ABTS radical was PG > HQ > MPBHQ > TBHQ > BHT. Different from the results of the DPPH radical scavenging assay, the ABTS radical scavenging activity of MPBHQ with four hydroxyl groups was higher than that of TBHQ with two hydroxyl groups and BHT with one hydroxyl group, but lower than that of PG with three hydroxyl groups and HQ with two hydroxyl groups. This may be due to the amphiphilic nature of benzothiazole structure of the ABTS radical rather than the hydrophobic aromatic ring structure of the DPPH radical, which makes it easier to undergo electron transfer or hydrogen transfer with phenolics [30,31]. Therefore, it is speculated that the alkyl groups attached to MPBHQ and TBHQ are the main reason for their lower scavenging effect on ABTS radical than PG and HQ. Overall, it can be demonstrated that MPBHQ has strong radical scavenging ability in non-oil systems, indicating its strong antioxidant ability. Whether MPBHQ can exhibit strong antioxidant activity in oil and fat with a more complex composition, it is necessary to carry out research on the antioxidant capacity of MPBHQ in oil and fat systems.

From the results of DPPH and ABTS assay (Figure 4a,b), MPBHQ exhibited strong single electron transfer ability for scavenging free radicals. However, it is generally believed that the method based on the HAT mechanism is similar to the classical lipid peroxidation method, which is closest to the real oxidation process of the physiological environment. Therefore, in order to detect the antioxidant activity of MPBHQ in lipids, ORAC assay was carried out.

### 3.4. ORAC Assay

The antioxidant capacity of phenolic compounds mainly relies on the hydroxyl group on their basic backbone to provide [H], thus achieving the purpose of scavenging free radicals and terminating free radical chain reactions. The substitution position and number of hydroxyl groups are important for their antioxidant activity. The antioxidant capacity of phenolics is enhanced with the degree of hydroxylation. However, the ORAC value of HQ is larger than both TBHQ with two hydroxyl groups and PG with three hydroxyl groups, which is attributed to the fact that HQ is more hydrophilic in comparison, which causes it to release [H] and then scavenge free radicals more easily in aqueous solutions. MPBHQ has side chain alkyl groups, but it also contains a higher number of hydroxyl groups. Thus, MPBHQ exhibited a higher ability to scavenge free radicals than that of TBHQ, PG and BHT. The ORAC assay responded to the ability of MPBHQ to provide [H] to convert free radicals into more stable products to stop free radical chain reactions. The results of SET (DPPH and ABTS assay) and HAT (ORAC assay) assay shown that MPBHQ has a stronger ability to transfer hydrogen and single electrons. This will make it have strong application potential in the food and chemical industry. Furthermore, we selected a lard system and mixed fatty acid methyl ester system to evaluate the value and effect of MPBHQ as an antioxidant in edible oil and biodiesel using the Rancimat test.

### 3.5. Antioxidant Activity in Lard and Fatty Acid Methyl Esters—Rancimat Test

The Rancimat test was used to comparatively study the antioxidant activity of BHT, HQ, TBHQ, PG, and MPBHQ in the pure oil system, and the antioxidant activity of MPBHQ was comprehensively evaluated. Lard without antioxidants was selected as raw material, and the antioxidant concentrations were set to 0.01, 0.02, and 0.04% (*w*/*w*). The execution temperature of Rancimat tests was set in the range of 120–140 °C, and the tests were carried out at air-saturated conditions. The maximum concentration of antioxidant compounds was set as 400 mg/kg to study whether the antioxidant activity of MPBHQ changed with increasing concentration. Additionally, the oxidation induction time (IP) and antioxidant protection factor (Pf) were used as evaluation indices. Generally, higher Pf values indicated stronger antioxidant activity. Specifically, Pf < 1 indicated pro-oxidant effect, Pf = 1 indicated no antioxidant activity, 1 < Pf < 2 showed weak antioxidant activity, 2 < Pf < 3 indicated effective antioxidant activity, and Pf > 3 showed that the compound had strong antioxidant activity [32,33]. According to the trend of Pf values shown in Figure 5a, when the addition amount of antioxidant compounds is 0.02%, the antioxidant capacity is MPBHQ > PG > TBHQ > HQ > BHT at 120, 130, and 140 °C. When the execution temperature is 120 °C, the Pf value of lard with MPBHQ is 1.4, 1.5, 2.5, and 4 times than that of lard with PG, TBHQ, HQ, and BHT, respectively. When the execution temperature rises to 140 °C, the Pf value of lard with MPBHQ is about 2.0, 2.0, 3.0, and 4.2 times than that of lard with PG, TBHQ, HQ, and BHT, respectively. This indicates that MPBHQ has better antioxidant capacity than that of PG, TBHQ, HQ, and BHT, and this may be mainly because MPBHQ has a higher relative molecular mass and evaporates less at relatively high temperatures [18]. Furthermore, the higher number of phenolic hydroxyl groups is the main reason for the strong antioxidant capability of MPBHQ, and the presence of the methyl propyl electron-giving substituent in the structure further enhances its antioxidant activity [19,34].

According to the trend of Pf values shown in Figure 5b, when the heating condition was 120 °C, all five antioxidant compounds showed stronger antioxidant performance with an increase in additive amount (0–0.04 wt.%). When the additive amount of MPBHQ was 0.01 wt.%, the Pf value was 4.90. However, the addition of PG or TBHQ should be doubled to achieve similar antioxidant effects, and their corresponding Pf values were 4.90 and 4.88, respectively. It is even less optimistic that when the addition amount was 0.04%, the Pf values of HQ and BHT were only 3.21 and 2.32, respectively. In general, for the oil system, MPBHQ has the strongest antioxidant capacity. Therefore, MPBHQ has a broad application prospect due to its strong antioxidant capacity and thermal stability, including the antioxidant protection of industrial fats, amongst others.

In order to further explore the performance of MPBHQ and broaden its application scenarios, its antioxidant capacity in mixed fatty acid methyl esters was evaluated. According to the trend of Pf values shown in Figure 5c, when the addition amount of antioxidant compounds was 0.02%, the antioxidant capacity was MPBHQ > PG > TBHQ > HQ > BHT at 120, 130 and 140 °C of fatty acid methyl esters. According to the trend of Pf values shown in Figure 5d, when the heating condition was 120 °C, all five antioxidant compounds showed stronger antioxidant performance with the increase in additive amount (0–0.04 wt.%). In summary, MPBHQ also exhibits excellent antioxidant properties in fatty acid methyl esters, which contain polyhydroxyl and alkane structures that give it both good oil solubility and good antioxidant properties. Fatty acid methyl esters are the main components of biodiesel, so MPBHQ can be used as a potential antioxidant for biodiesel.

Table 2 shows the antioxidant properties of some synthetic antioxidants compared with MPBHQ. MPBHQ has the strongest advantage in DPPH free radical scavenging tests. In the Rancimat test, MPBHQ showed a strong antioxidant effect, but was second to 6,6′-(butane-1,1-diyl)bis(4-methylbenzene-1,2-diol) (BMB) studied by Olajide et al. [18]. In summary, MPBHQ has strong free radical scavenging ability and prevents oxidative rancidity of oil and fat.

## 4. Conclusions

In this work, a novel antioxidant named MPBHQ containing four phenolic hydroxyl groups was successfully synthesized. After optimized separation and purification, the chemical structure of MPBHQ was identified, and its antioxidant properties were evaluated by free radical scavenging assays based on the SET mechanism and HAT mechanism, as well as the oxidative rancidity tests of a lard system and a mixed fatty acid methyl ester system. The chemical structure of MPBHQ was confirmed by the HRMS, NMR, FTIR and UV spectroscopy, and its chemical name was designated as 2,2′-(2-methylpropane-1,3-diyl)bis(hydroquinone). The results of DPPH and ABST radical scavenging assay indicate that MPBHQ has an excellent ability to scavenge free radicals dominated by SET mechanism. MPBHQ also showed strong antioxidant properties in the ORAC assay, and this indicated that MPBHQ also had excellent hydrogen transfer ability. Furthermore, the results of Rancimat test show that MPBHQ has excellent ability to inhibit the oxidation of edible oil and mixed fatty acid methyl ester. By comparing the oxidation resistance test results of MPBHQ, PG, TBHQ, HQ, and BHT, the reason why MPBHQ has higher antioxidant capacity is mainly because it has more phenolic hydroxyl groups and the methyl propyl electron-giving substituent, as well as the relatively higher thermostability. In summary, MPBHQ meets the requirements of a new antioxidant with strong antioxidant properties at high temperatures, and MPBHQ will have great potential application value in industrial products such as biodiesel and edible oils.

## Data Availability

Data are contained within the article.

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
