# Peer review of "Synthesis, Characterization and Evaluation of a Novel Tetraphenolic Compound as a Potential Antioxidant"

_antioxidants, 2023, doi:10.3390/antiox12071473_

Round 1

Reviewer 1 Report

The manuscript is quite interesting and the purpose of this study is clearly described. However, some parts are not clear and needs to be clarified.

-The structure elucidation is not sufficient. Did you check HMBC? How did you confirm the correlations?

- The antioxidant activity is described repeatedly. Please make it clear and concise.

-The antioxidant was measured by several assay system. Discussion about each method and results should be added.

- What do the a,b,c,d mean in Figures?

- There are so many typos in the manuscript. ex) superscript, italic etc.

The results are interesting but the manuscript can be improved by more discussion and careful revision.

Many typos throughout the manuscript.

Author Response

Dear Editor and Reviewers,

Thank you for your letter and for the reviewers’ comments concerning our manuscript entitled “Synthesis, characterization and evaluation of a novel tetraphenolic compounds as potential antioxidants” (Ms. Ref. No.: antioxidants-2465581). The comments or suggestions raised by the reviewers are very professional and pertinent. Those comments are all valuable and very helpful for revising and improving our manuscript. We have studied comments carefully and have made correction which we hope to meet with approval. Revised portions are marked in red colored text in our revised manuscript. We appreciate the hard work and the great contributions of the reviewers and the editors for this manuscript. The main corrections in the manuscript and the responses to the reviewers’ comments (point-to-point) are as follows:

To Reviewer #1:

Thank you for your letter. And, special thanks to you for your good comments. The comments or suggestions raised are very professional and pertinent. The original manuscript was changed and marked in red colored text in the revised manuscript.
1. The structure elucidation is not sufficient. Did you check HMBC? How did you confirm the correlations?

Re: Thanks for your comment. In the revised manuscript, we used TLC (Rf value) and ultraviolet spectroscopy to identify the structure species of target compound. Also, FTIR, HRMS (ESI) and NMR was used to analyze the detailed structure information of the compounds and confirmed the chemical structure of MPBHQ. Based on the 1H-NMR (Figure 3a), 13C-NMR (Figure 3b), 1H-1H COSY-NMR (Figure 3c) and 13C-1H HSQC-NMR (Figure 3d), we checked the HMBC spectra of MPBHQ.

Test conditions: (L169-171) "2D HMBC spectra were acquired with the pulse program “hmbcgpndqf” with a relaxation delay of 1.5 s and the J(XH) constant of 8 Hz. A data matrix of 1024 × 256 points was recorded with 8 scans for each increment. " The results and discussions are as follows.

(L283-305) The chemical structure of MPBHQ was identified by NMR spectra. The NMR spectra of MPBHQ was shown in (Figure 3). As shown in Figure 3a, chemical shifts of hydrogen on the benzene ring were located at around 6-7 ppm. The hydrogen on C9 feels the biggest electron shielding effect in the molecule structure, so it shows the smallest chemical shift in the high field region at 0.90 ppm. The chemical shifts of hydrogen at H4 and H5 are 3.98 ppm and 2.52 ppm, respectively. All the assignments of hydrogen chemical shifts in Figure 3a were further determined by the COSY spectra (Figure 3c). Information on the coupling of each group is obtained from the relevant peaks outside the diagonal of the COSY spectrum. We can easily get the coupling information of H5 and H6 for H4 and H5. Based the correlation signals of C-H in HSQC spectra (Figure 3d), all the assignments of carbons in MPBHQ molecules were shown in Figure 3b. The chemical shifts of carbons connecting the two phenolic hydroxyl positions (C1 and C4) are determined at 150.9 ppm and 149.5 ppm, respectively. Chemical shifts of carbons attached to hydrogens on the benzene ring were determined at the range of 110-120 ppm. For the alkyl fraction, the chemical shift of the methyl carbon was determined at 23.6 ppm. The C8 of methylene was assigned to the chemical shift of 33.6 ppm. Then from Figure 3d we found the correlation between the three hydrogen atoms on the benzene ring except the hydrogen atom on the hydroxyl group and the carbon atom of the benzene ring, which are H1, H2, H3. The correlations of hydrogen atoms H4, H5, H6 on alkyl corresponding to C7, C8, C9 are also marked in the figure. In Figure 3e, the correlations of H and C in adjacent or interphase positions are marked, such as the correlations of H6 and C7, H1and C5. In summary, the chemical information of MPBHQ was determined, and it is an alkylbenzene compound containing four hydroxyl groups, as shown in Figure 3.

  1. The antioxidant activity is described repeatedly. Please make it clear and concise.

Re: The reviewer’s suggestion has been sufficiently considered. According to the reviewers' comments, we supplemented the oxygen radical absorption capacity (ORAC) assay and modified the discussion related to the free radical scavenging test. We hope that this modification can more fully reflect the antioxidant activity of MPBHQ.

Based on the selected method in this work, the DPPH radical scavenging assay in methanol solution is mainly dominated by single electron transfer (SET) mechanism. However, ABTS radicals can be eliminated by both SET and hydrogen atom transfer (HAT) mechanism. In order to evaluate the antioxidant activity of MPBHQ in detail, ABTS radical scavenging assay was carried out. From the results of DPPH and ABTS assay, MPBHQ exhibited strong single electron transfer ability for scavenging free radicals. However, it is generally believed that the method based on the HAT mechanism is similar to the classical lipid peroxidation method, which is closest to the real oxidation process of the physiological environment. Therefore, in order to detect the antioxidant activity of MPBHQ in lipids, ORAC assay was carried out. The ORAC assay responded to the ability of MPBHQ to provide [H] to convert free radicals into more stable products to stop free radical chain reactions. The results of SET (DPPH and ABTS assay) and HAT (ORAC assay) assay shown that MPBHQ has a stronger ability to transfer hydrogen and single electrons. This will make it have strong application potential in the food and chemical industry.

The detailed modification was shown in Line 337 to Line 341 and Line 361 to Line 384 of the revised manuscript.

  1. The antioxidant was measured by several assay system. Discussion about each method and results should be added.

Re: Thanks for your comment. The reviewer’s suggestion has been sufficiently considered. We have revised the relevant content, mainly expounding the logical relationship of several assay system. We explained the pattern of antioxidant activity in terms of SET (DPPH and ABTS assay) and HAT (ORAC assay) mechanisms, respectively. Also, we added the Rancimat test in the fatty acid methyl ester system. The detailed modification was shown in Line 337 to Line 341, Line 361 to Line 384 and Line 428 to Line 446 of the revised manuscript and highlighted in red font. The newly added figures and discussions in the revised manuscript are as follows.

Figure 4(c) The attenuation curve of fluorescence intensity of Trolox solution, (d) ORAC values of different antioxidants.

(Line 368-384) The antioxidant capacity of phenolic compounds mainly relies on the hydroxyl group on their basic backbone to provide [H], thus achieving the purpose of scavenging free radicals and terminating free radical chain reactions. The substitution position and number of hydroxyl groups are important for their antioxidant activity. The antioxidant capacity of phenolics is enhanced with the degree of hydroxylation. However, the ORAC value of HQ is larger than both TBHQ with two hydroxyl groups and PG with three hydroxyl groups, which is attributed to the fact that HQ is more hydrophilic in comparison, which causes it to release [H] and scavenge free radicals more easily in aqueous solutions. MPBHQ has side chain alkyl groups, but it also contains a higher number of hydroxyl groups. So, MPBHQ exhibited a higher ability to scavenge free radicals than that of TBHQ, PG and BHT. The ORAC assay responded to the ability of MPBHQ to provide [H] to convert free radicals into more stable products to stop free radical chain reactions. The results of SET (DPPH and ABTS assay) and HAT (ORAC assay) assay shown that MPBHQ has a stronger ability to transfer hydrogen and single electrons. This will make it have strong application potential in the food and chemical industry. Furthermore, we selected lard system and mixed fatty acid methyl ester system to evaluate the value and effect of MPBHQ as an antioxidant in edible oil and bio-diesel using the Rancimat test.

In order to further explore the performance of MPBHQ and broaden its application scenarios, its antioxidant capacity in mixed fatty acid methyl esters was evaluated. The results and discussions were shown in Line 435 to Line 446 of the revised manuscript. The relevant figures and discusses are as follows.

Figure 5(c) Changes in Pf values of fatty acid methyl ester containing 0.02 wt.% of antioxidants at different temperatures, (d) Changes in Pf values of fatty acid methyl ester containing different antioxidants concentrations at 120°C.

(Line 435-446) In order to further explore the performance of MPBHQ and broaden its application scenarios, its antioxidant capacity in mixed fatty acid methyl ester was evaluated. According to the trend of Pf values shown in Figure 5c, when the addition amount of antioxidant compounds is 0.02%, the antioxidant capacity is MPBHQ > PG > TBHQ > HQ > BHT at 120, 130 and 140°C of fatty acid methyl esters. According to the trend of Pf values shown in Figure 5d, when the heating condition was 120°C, all the five antioxidant compounds showed stronger antioxidant performance with the increase of additive amount (0-0.04 wt.%). In summary, MPBHQ also exhibits excellent antioxidant properties in fatty acid methyl esters, which contain polyhydroxyl and alkane structures that give it both good oil solubility and good antioxidant properties. Fatty acid methyl esters are the main components of biodiesel, so MPBHQ can be used as a potential antioxidant for biodiesel.

  1. What do the a, b, c, d mean in Figures?

Re: Thanks for your comment. The a, b, c, d, e, and f represent the significant differences and were displayed in the figure captions.

(L388-390) " The different lowercase letters in the graph, including a, b, c, d, e and f, indicate the significant difference of EC50 values of the same substance at different concentrations (P<0.05)."

(L422-424) " The different lowercase letters in the graph, including a, b, c, d, e and f, indicate the significant difference of Pf values of the different substance at same condition (P<0.05)."

  1. There are so many typos in the manuscript. ex) superscript, italic etc.

Re: The reviewer’s suggestion has been sufficiently considered. We are sorry for this error and have made corrections in the revised manuscript.

(L324-325) "EC50 values of TBHQ, HQ, PG and MPBHQ were 22.20, 11.34, 8.74 and 7.93 μg/mL, respectively. Because of the weak ability of BHT, the EC50 values of BHT cannot be calculated. "

(L345-346)"EC50 values of TBHQ, MPBHQ, HQ and PG were 33.34, 24.35, 21.81 and 18.17 μg/mL, respectively. "

Yours sincerely,

Yan-Lan Bi

School of Food Science and Engineering

Henan University of Technology

Zhengzhou 450001

  1. R. China

Tel: (086)371-67758022,

Fax: (086)371-67758022,

E-mail: byl@haut.edu.cn.

Reviewer 2 Report

The manuscript by Bi and coworkers describes the synthesis of 2,2'-(2-methylpropane-1,3-diyl)bis(hydroquinone) (MPBHQ) and a brief characterization of its antioxidant profile. While the results are interesting, they are only preliminary. A more extensive library should be synthesized, more varied antioxidant characterization methods should be employed (e.g. ORAC) and applications should also be expanded. In summary, I do not think that the amount and quality of this work is enough to merit publication in a high-impact journal such as Antioxidants. 

No comments

Author Response

Dear Editor and Reviewers,

Thank you for your letter and for the reviewers’ comments concerning our manuscript entitled “Synthesis, characterization and evaluation of a novel tetraphenolic compounds as potential antioxidants” (Ms. Ref. No.: antioxidants-2465581). The comments or suggestions raised by the reviewers are very professional and pertinent. Those comments are all valuable and very helpful for revising and improving our manuscript. We have studied comments carefully and have made correction which we hope to meet with approval. Revised portions are marked in red colored text in our revised manuscript. We appreciate the hard work and the great contributions of the reviewers and the editors for this manuscript. The main corrections in the manuscript and the responses to the reviewers’ comments (point-to-point) are as follows:

To Reviewer #2:

Thank you for your letter and patience. And, special thanks to you for your good comments. The comments or suggestions raised are very professional and pertinent. We appreciated and sufficiently considered the reviewer’s comments, and the original manuscript was changed and marked in red colored text in the revised manuscript.
1. A more extensive library should be synthesized more varied antioxidant characterization methods should be employed (e. g. ORAC) and applications should also be expanded.

Re: Thanks for your comment. The reviewer’s suggestion has been sufficiently considered. We have added the antioxidant characterization methods and applications of compound. For this work, our main purpose is to be able to synthesize and purify antioxidants with multiple hydroxyl groups and to be able to verify that multiple hydroxyl groups and alkanes in the structure can promote their antioxidant capacity in oils and fats, etc. Therefore, in this manuscript, we studied the main conditions affecting the synthesis and focused on the antioxidant properties. A purposefully designed catalytic pathway was selected to synthesize the target compound, and the novel potential antioxidant containing multiple hydroxyl groups and alkanes was separated and purified. The obtained substance that meets the target requirements and verified the rationality of the theory with an efficient experimental design. Because the synthesis path in this work is a goal-directed selective route, a more detailed experimental design is underway for more efficient synthesis of MPBHQ.

Then, for the varied antioxidant characterization methods, we added the ORAC test and the Rancimat test in fatty acid methyl esters. MPBHQ also exhibits excellent antioxidant properties in fatty acid methyl esters, which are the main components of biodiesel, so MPBHQ can also be used as a potential antioxidant for biodiesel. The related results and discussion are as follow.

(L212-224) The ORAC assay was measured by modification of a previous method. This test method used an enzyme marker for an excitation wavelength of 485 nm and an emission wavelength of 530 nm. In each well, 150 μL of FL (78 nM) and 25 μL of sample, blank (PBS), or standard (Trolox, 6.25-100 μmol/L) were placed, and then 25 μL of AAPH (73 mM) were added. The fluorescence was measured immediately after the addition and measurements were then taken every 5 min until the relative fluorescence intensity (FI%) was less than 5% of the value of the initial reading. The measurements were taken in triplicate. The area under the fluorescence decay curve (AUC) was calculated by applying the following formula:

ORAC (μm TE)=(CTrolox/Csample)·[(AUCsample-AUCblank)/(AUCTrolox-AUCblank)]

AUC=0.5×[2×(f0+f1+…+fn-1+fn)-f0-fn]â–³t

Where f0 is the initial fluorescence and fn is the fluorescence at time n.

Figure 4(c) The attenuation curve of fluorescence intensity of Trolox solution, (d) ORAC values of different antioxidants.

(L368-384) The antioxidant capacity of phenolic compounds mainly relies on the hydroxyl group on their basic backbone to provide [H], thus achieving the purpose of scavenging free radicals and terminating free radical chain reactions. The substitution position and number of hydroxyl groups are important for their antioxidant activity. The antioxidant capacity of phenolics is enhanced with the degree of hydroxylation. However, the ORAC value of HQ is larger than both TBHQ with two hydroxyl groups and PG with three hydroxyl groups, which is attributed to the fact that HQ has no side chain groups and is more hydrophilic in comparison, which causes it to release [H] and scavenge free radicals more easily in aqueous solutions. MPBHQ has side chain alkyl groups, but it contains a higher number of hydroxyl groups and has a higher ability to scavenge free radicals. The ORAC assay responded to the ability of MPBHQ to provide [H] to convert free radicals into more stable products to stop free radical chain reactions.

Figure 5(c) Changes in Pf values of fatty acid methyl ester containing 0.02 wt.% of antioxidants at different temperatures, (d) Changes in Pf values of fatty acid methyl ester containing different antioxidants concentrations at 120°C.

(L436-446) According to the trend of Pf values shown in Figure 5c, when the addition amount of antioxidant compounds is 0.02%, the antioxidant capacity is MPBHQ > PG > TBHQ > HQ > BHT at 120, 130 and 140℃ of fatty acid methyl. According to the trend of Pf values shown in Figure 5d, when the heating condition was 120℃, all the five antioxidant compounds showed stronger antioxidant performance with the increase of additive amount (0-0.04 wt.%). MPBHQ also exhibits excellent antioxidant properties in fatty acid methyl esters, which contain polyhydroxyl and alkane structures that give it both good oil solubility and good antioxidant properties. Fatty acid methyl esters are the main components of biodiesel, so MPBHQ can be used as a potential antioxidant for biodiesel.

Yours sincerely,

Yan-Lan Bi

School of Food Science and Engineering

Henan University of Technology

Zhengzhou 450001

  1. R. China

Tel: (086)371-67758022,

Fax: (086)371-67758022,

E-mail: byl@haut.edu.cn.

Reviewer 3 Report

The paper aimed to study the antioxidant activity of a novel synthetic phenolic compound by DPPH and ABTS radical scavenging assays and the Rancimat test.

The idea of the work seems to be interesting, especially since it is so important to search for new stable antioxidant agents. The manuscript is well-written, and the conclusions are consistent with the results obtained. However, the manuscript has two major flaws. Firstly, the experimental part lacks some data important for the repetition of the work and verification of its scientific soundness. Secondly, the work lacks proper discussion – the obtained results are simply described with no mention of any possible implications and impact of the experimental observations. I think the submission requires extensive edition and improvement at many points, the main of which are listed below.

Major issues:

The first drawback of the work is the very small number of antioxidant activity tests in which the analysis of the newly synthesized compound was performed. In this work, the antioxidant activity of MPBHQ was tested only in three simple cell-free in-vitro tests, two SET-based tests (DPPH and ABTS) and the Rancimat test. It is known that antioxidant tests reacting in the HAT mechanism better reflect the in vivo conditions of the human body. For the Authors to conclude that: “MPBHQ … will have broad application prospects in the food and chemical industries” (L85), additional in vitro analyses of antioxidant activity in tests reacting in the HAT mechanism (e.g. inhibition of linoleic acid oxidation) or tests measuring the ability to scavenge important in vivo radicals such as hydroxyl radical or superoxide anion radical must be performed and the results added to the manuscript.

The second major issue is the lack of any discussion. The obtained results are simply described with no mention of any possible implications and impact of the experimental observations. At this point, it is necessary to add a critical discussion of the antioxidant activity obtained by the Authors for the newly synthesized compound in various cell-free in vitro tests and to compare these results with reports on other synthetic antioxidant agents containing multiple hydroxyl groups.

Minor issues:

L33, 54, etc.: please add a break in the text (for example: [10]by). Please check the whole manuscript thoroughly.

L126-128: the gradient profile for HPLC-PDA assay should be given using percentages of solvent A. For example: “The elution profile was as follows: 0-1 min, 5% A (v/v); 1-8 min, 5-30% A; 8-12 min, 30-95% A, etc.”.

L152, 154, 211, 233, etc.: Authors should decide whether to use gap (180 °C) or continuous (180°C) temperature notation and should use this consistently throughout the manuscript.

L209: please eliminate unnecessary space in the text of the manuscript. Please check the whole manuscript thoroughly.

L165-167: interpretation of 1H NMR and 13C NMR spectra should include precise assignment of signals to specific H and C atoms, e.g.: “1H NMR δ 6.51 (1H, d, J = 1.9 Hz, H-2)” or 13C NMR δ 118.2 (C-2)”. At this point, please also add a Figure containing the chemical formula of the compound with numbered carbon atoms, so that the NMR analysis can be interpreted more easily. Figures 3a and 3b contain such chemical structures, but these figures have too low resolution and are therefore illegible.

L177-181: this description is redundant. The method of determining the retention factor (Rf) is well-known in the world of science.

L248: The UV absorption peak at 205 nm is not visible in Figure 1b, because the wavelength range (nm) only starts at 250nm.

L255: please remove unnecessary dot (vibrations. of C-H).

L279-281: Figures 3a-d have too low a resolution and are therefore illegible. Please insert larger figures with higher resolution, preferably in the Supplements section.

L294, 295, 311, etc.: please correct the notation EC50 with the number 50 as a subscript. Please check the whole manuscript thoroughly.

Author Response

Dear Editor and Reviewers,

Thank you for your letter and for the reviewers’ comments concerning our manuscript entitled “Synthesis, characterization and evaluation of a novel tetraphenolic compounds as potential antioxidants” (Ms. Ref. No.: antioxidants-2465581). The comments or suggestions raised by the reviewers are very professional and pertinent. Those comments are all valuable and very helpful for revising and improving our manuscript. We have studied comments carefully and have made correction which we hope to meet with approval. Revised portions are marked in red colored text in our revised manuscript. We appreciate the hard work and the great contributions of the reviewers and the editors for this manuscript. The main corrections in the manuscript and the responses to the reviewers’ comments (point-to-point) are as follows:

To Reviewer #3:

Thank you for your letter and patience. And, special thanks to you for your good comments. The comments or suggestions raised are very professional and pertinent. We appreciated and sufficiently considered the reviewer’s comments, and the original manuscript was changed and marked in red colored text in the revised manuscript.

  1. The first drawback of the work is the very small number of antioxidant activity tests in which the analysis of the newly synthesized compound was performed. In this work, the antioxidant activity of MPBHQ was tested only in three simple cell-free in-vitro tests, two SET-based tests (DPPH and ABTS) and the Rancimat test. It is known that antioxidant tests reacting in the HAT mechanism better reflect the in vivo conditions of the human body. For the Authors to conclude that: “MPBHQ … will have broad application prospects in the food and chemical industries” (L85), additional in vitro analyses of antioxidant activity in tests reacting in the HAT mechanism (e.g. inhibition of linoleic acid oxidation) or tests measuring the ability to scavenge important in vivo radicals such as hydroxyl radical or superoxide anion radical must be performed and the results added to the manuscript

Re: Thanks for your comment. The reviewer’s suggestion has been sufficiently considered. We have added the antioxidant characterization methods and applications of compound. We explained the pattern of antioxidant activity in terms of SET (DPPH and ABTS assay) and HAT (ORAC assay) mechanisms, respectively. Also, we added the Rancimat test in the fatty acid methyl ester system. The results of ORAC assay were shown in Figure 4c and 4d of the revised manuscript, and the results of the Rancimat test in the fatty acid methyl ester system were shown in Figure 5c and 5d of the revised manuscript.

Figure 4(c) The attenuation curve of fluorescence intensity of Trolox solution, (d) ORAC values of different antioxidants.

(L368-384) The antioxidant capacity of phenolic compounds mainly relies on the hydroxyl group on their basic backbone to provide [H], thus achieving the purpose of scavenging free radicals and terminating free radical chain reactions. The substitution position and number of hydroxyl groups are important for their antioxidant activity. The antioxidant capacity of phenolics is enhanced with the degree of hydroxylation. However, the ORAC value of HQ is larger than both TBHQ with two hydroxyl groups and PG with three hydroxyl groups, which is attributed to the fact that HQ has no side chain groups and is more hydrophilic in comparison, which causes it to release [H] and scavenge free radicals more easily in aqueous solutions. MPBHQ has side chain alkyl groups, but it contains a higher number of hydroxyl groups and has a higher ability to scavenge free radicals. The ORAC assay responded to the ability of MPBHQ to provide [H] to convert free radicals into more stable products to stop free radical chain reactions.

Figure 5(c) Changes in Pf values of fatty acid methyl ester containing 0.02 wt.% of antioxidants at different temperatures, (d) Changes in Pf values of fatty acid methyl ester containing different antioxidants concentrations at 120°C.

(L436-446) According to the trend of Pf values shown in (Figure 5c), when the addition amount of antioxidant compounds is 0.02%, the antioxidant capacity is MPBHQ > PG > TBHQ > HQ > BHT at 120, 130 and 140℃ of fatty acid methyl. According to the trend of Pf values shown in (Figure 5d), when the heating condition was 120℃, all the five antioxidant compounds showed stronger antioxidant performance with the increase of additive amount (0-0.04 wt.%). MPBHQ also exhibits excellent antioxidant properties in fatty acid methyl esters, which contain polyhydroxyl and alkane structures that give it both good oil solubility and good antioxidant properties. Fatty acid methyl esters are the main components of biodiesel, so MPBHQ can be used as a potential antioxidant for biodiesel.

  1. The second major issue is the lack of any discussion. The obtained results are simply described with no mention of any possible implications and impact of the experimental observations. At this point, it is necessary to add a critical discussion of the antioxidant activity obtained by the Authors for the newly synthesized compound in various cell-free in vitro tests and to compare these results with reports on other synthetic antioxidant agents containing multiple hydroxyl groups.

Re: Thanks for your comment. In the revised manuscript, based on the various cell-free in vitro tests and the SET/HAT mechanism, we summarized the results of each test and elaborated the logical relationship of various cell-free in vitro tests, hoping to help readers better understand our work. Moreover, we compared the performance of MPBHQ with other other synthetic antioxidant agents containing multiple hydroxyl groups. Also, we added the Rancimat test in the fatty acid methyl ester system. The detailed modification was shown in Line 337 to Line 341, Line 361 to Line 384 and Line 428 to Line 446 of the revised manuscript and highlighted in red font.

Based on the selected method in this work, the DPPH radical scavenging assay in methanol solution is mainly dominated by single electron transfer (SET) mechanism. However, ABTS radicals can be eliminated by both SET and hydrogen atom transfer (HAT) mechanism. In order to evaluate the antioxidant activity of MPBHQ in detail, ABTS radical scavenging assay was carried out. From the results of DPPH and ABTS assay, MPBHQ exhibited strong single electron transfer ability for scavenging free radicals. However, it is generally believed that the method based on the HAT mechanism is similar to the classical lipid peroxidation method, which is closest to the real oxidation process of the physiological environment. Therefore, in order to detect the antioxidant activity of MPBHQ in lipids, ORAC assay was carried out. The ORAC assay responded to the ability of MPBHQ to provide [H] to convert free radicals into more stable products to stop free radical chain reactions. The results of SET (DPPH and ABTS assay) and HAT (ORAC assay) assay shown that MPBHQ has a stronger ability to transfer hydrogen and single electrons. This will make it have strong application potential in the food and chemical industry.

Table 2 Comparison of the antioxidant properties of MPBHQ and several synthetic polyhydroxy antioxidants

Antioxidant

Structure

Condition of DPPH

 EC50/μg·mL-1

Condition of Rancimat

Pf

reference

Compound 1

0.5mL of sample, 2.5mL of DPPH (0.1 mM)

31.64

3 g of Lard, 120℃, 0.02% of antioxidant, 20 L/h of air flow rate

6.12

[19]

Compound 2

0.5mL of sample, 2.5mL of DPPH (0.1 mM)

106.95

3 g of Lard, 120℃, 0.02% of antioxidant, 20 L/h of air flow rate

7.28

[19]

6,6'-(butane-1,1-diyl)

bis(4-methylbenzene-1,2-diol)

0.5mL of sample, 3mL of DPPH (0.1 mM)

24.39

3 g of Lard, 120℃, 0.02% of antioxidant, 20 L/h of air flow rate

14.85

[18]

MPBHQ

0.5mL of sample, 3mL of DPPH (0.1 mM)

7.93

3 g of Lard, 120℃, 0.02% of antioxidant, 20 L/h of air flow rate

7.25

This work

TBHQ

0.5mL of sample, 3mL of DPPH (0.1 mM)

22.20

3 g of Lard, 120℃, 0.02% of antioxidant, 20 L/h of air flow rate

4.88

This work

PG

0.5mL of sample, 3mL of DPPH (0.1 mM)

11.34

3 g of Lard, 120℃, 0.02% of antioxidant, 20 L/h of air flow rate

4.90

This work

(L450-454) The table 2 shows the antioxidant properties of some synthetic antioxidants compared with MPBHQ. MPBHQ has a strongest advantage in DPPH free radical scavenging tests. In the Rancimat test, MPBHQ showed a strong antioxidant effect, but was second to BMB studied by Olajide et al [18]. In summary, MPBHQ has strong free radical scavenging ability and prevents oxidative rancidity of oils and fats.

  1. L33 54, etc.: please add a break in the text (for example: [10]by). Please check the whole manuscript thoroughly.

Re: Thanks for your comment. We are sorry for this error and have made corrections in the revised manuscript. (L31-34)" The mainstream of the market is still synthetic phenolic antioxidants, which are widely used in food [1], medicine [6], cosmetics [7], fuel [8], feed [9] and other fields [10] by virtue of their low price, high yield and good antioxidant effect." (L53-55) "Hydroquinone (HQ) is an important raw material additive and intermediate in the synthesis of pharmaceuticals [20] and other fine chemicals. But HQ is less used in the oils and fats processing industry due to its poor oil solubility, small molecular weight and easy volatility [21]. "

  1. L126-128: the gradient profile for HPLC-PDA assay should be given using percentages of solvent A. For example. "The elution profile was as follows: 0-1 min, 5% A (); 1-8 min, 5-30% A; 8-12min, 30-95%Aetc."

Re: Thanks for your comment. We have made modifications in the revised manuscript. (L129-132)"The elution profile was as follows: 0 min, 60% A (aqueous solution containing 0.5% acetic acid); 1-8 min, 60-20% A; 8-15 min, 20-0% A; 15-20 min, 0% A; 20-28 min, 0-60% A. "

  1. L152, 154, 211, 233, etc.: Authors should decide whether to use gap (180℃) or continuous(180°C) temperature notation and should use this consistently throughout the manuscript

Re: Thanks for your comment. We are sorry for this error and have made corrections in the revised manuscript. (L156-157)"The ToF MS Functions were set as follows. Desolvation temperature was 180°C, source temperature was 80°C." (L253-254) " When the molar ratio of hydroquinone, methylallyl alcohol and H3PO4 was 1:1.2:0.5, the reaction temperature was 90°C, and the reaction time was 20 min, the content of MPBHQ reached 41%."

  1. L209: please eliminate unnecessary space in the text of the manuscript Please check the whole manuscript thoroughly.

Re: Thanks for your comment. We are sorry for this error and have made corrections in the revised manuscript. (L226-230)"The antioxidant activities of MPBHQ, BHT, HQ, TBHQ and PG in lard were determined according to AOCS Official Method Cd 12b-92 Oil Stability Index, and the data was collected on a Rancimat 892 (Metrohm, Herisau, Switzerland). 3.00±0.01 g of lard containing antioxidant were subjected to accelerated oxidation under the following conditions. "

  1. L165-167: interpretation of 1H NMR and 13C NMR spectra should include precise assignment of signals to specific H and C atoms, eg.: “H NMR 6.55 (1H, d, J = 1.9 HH- or 13C NMR 118.2 (C-2)". At this point, please also add a Figure containing the chemical formula of the compound with numbered carbon atoms so that the NMR analysis can be interpreted more easily. Figures 3a and 3b contain such chemical structures, but these figures have too low resolution and are therefore illegible

Re: Thanks for your comment. We are sorry for this error and have made corrections in the revised manuscript. (L171-175) "1H NMR (500 MHz, Methyl alcohol-d) δ 6.55 (d, J = 1.9 Hz, 1H, H-3), 6.35 (d, J = 2.0 Hz, 1H, H-1), 0.90 (d, 3H, H-6), 2.52 (m, H, H-5), 3.98ppm (d, 2H, H-4). 13C NMR (500 MHz, Methyl alcohol-d) δ (118.2, C-5), δ (116.4, C-3), δ (115.2, C-2), δ (23.6, C-9), δ (33.6, C-8), δ (46.2, C-7)."

  1. L177-181: this description is redundant. The method of determining the retention factor (Rf) is well-known in the world of science.

Re: Thanks for your comment. We are sorry for this error and have made corrections in the revised manuscript. (L178-184)"After the expansion is completed, the solvent was evaporated, and iodine was used to develop the color. The Rf value was calculated based on the shift distance of sample and using the equation: Rf=dc/do"

  1. L248: The UV absorption peak at 205 nm is not visible in Figure 1b, because the wave length range (nm) only starts at 250 nm.

Re: Thanks for your comment. We are sorry for this error and have made corrections in the revised manuscript. (L269-271)"MPBHQ showed a strong UV absorption peak 297 nm. After mixing MPBHQ with KOH solution, the absorption peak is enhanced and red-shifted"

  1. L255: please remove unnecessary dot (vibrations. of C-H)

Re: Thanks for your comment. We are sorry for this error and have made corrections in the revised manuscript. (L276-278)" The peaks centered at 2947.88, 2834.50 cm-1 are attributed to the stretching vibration of C-H on alkanes, and 1422.41 cm-1 is attributed to the bending vibrations of C-H on alkanes."

  1. L279-281: Figures 3a-d have too low a resolution and are therefore illegible. Please insert larger figures with higher resolution, preferably in the Supplements section.

Re: Thanks for your comment. In the manuscript, a larger figure with higher resolution was inserted. Thank you again for your suggestion, we hope this modification can help readers clearly obtain the information of this work.

  1. L294, 295, 311, etc.: please correct the notation EC50 with the number 50 as a subscript. Please check the whole manuscript thoroughly.

Re: Thanks for your comment. We are sorry for this error and have made corrections in the revised manuscript." (L324-325) "EC50 values of TBHQ, HQ, PG and MPBHQ were 22.20, 11.34, 8.74 and 7.93 μg/mL, respectively. Because of the weak ability of BHT, the EC50 values of BHT cannot be calculated. " (L345-346)"EC50 values of TBHQ, MPBHQ, HQ and PG were 33.34, 24.35, 21.81 and 18.17 μg/mL, respectively. "

Yours sincerely,

Yan-Lan Bi

School of Food Science and Engineering

Henan University of Technology

Zhengzhou 450001

  1. R. China

Tel: (086)371-67758022,

Fax: (086)371-67758022,

E-mail: byl@haut.edu.cn.

Round 2

Reviewer 2 Report

The authors have suitably modified their manuscript, which can now be accepted for publication.

There are no significant issues with the quality of the English.

Reviewer 3 Report

The manuscript has been revised in accordance with the Reviewers' recommendations and is now suitable for publication in Antioxidants.